# The Frequency of Clinical Seizures in Paroxysmal Events in a Neonatal Intensive Care Unit

**DOI:** 10.3390/children9020238

**Published:** 2022-02-11

**Authors:** Oi-Wa Chan, Wan-Hsuan Chen, Jainn-Jim Lin, Ming-Chou Chiang, Shao-Hsuan Hsia, Huei-Shyong Wang, En-Pei Lee, Yi-Shan Wang, Cheng-Yen Kuo, Kuang-Lin Lin

**Affiliations:** 1Division of Pediatric Critical Care and Pediatric Neurocritical Care Center, Chang Gung Children’s Hospital and Chang Gung Memorial Hospital, Chang Gung University College of Medicine, Taoyuan 333, Taiwan; ai3333@cgmh.org.tw (O.-W.C.); tw1picu@gmail.com (S.-H.H.); pilichrislnp@gmail.com (E.-P.L.); 2Department of Pediatrics, Chiayi Chang Gung Memorial Hospital and Chang Gung University College of Medicine, Chiayi 613, Taiwan; wererabbit122@hotmail.com; 3Division of Pediatric Neurology, Chang Gung Children’s Hospital and Chang Gung Memorial Hospital, Chang Gung University College of Medicine, Taoyuan 333, Taiwan; wanghs444@cgmh.org.tw (H.-S.W.); hermitage513@gmail.com (Y.-S.W.); cykuo2000@hotmail.com (C.-Y.K.); lincgh@cgmh.org.tw (K.-L.L.); 4Department of Respiratory Therapy, Chang Gung Children’s Hospital and Chang Gung Memorial Hospital, Chang Gung University College of Medicine, Taoyuan 333, Taiwan; 5Division of Neonatology, Chang Gung Children’s Hospital and Chang Gung Memorial Hospital, Chang Gung University College of Medicine, Taoyuan 333, Taiwan; newborntw@gmail.com; 6Study Group for Intensive and Integrated Care of Pediatric Central Nervous System, Chang Gung Children’s Hospital, Taoyuan 333, Taiwan; bread86@cgmh.org.tw

**Keywords:** frequency, clinical seizure, paroxysmal events, NICU

## Abstract

Background: In general clinical practice, neonatal seizures are identified visually by direct clinical observation. The study aimed to examine the frequency of clinical seizures in paroxysmal events in a neonatal intensive care unit. Methods: We conducted a prospective study of continuous video-EEG monitoring in a neonatal intensive care unit between January 2017 and December 2020. The demographic data were also reviewed. Results: Sixty-four neonates were enrolled. The median total video-EEG monitoring duration was 24.1 h (IQR 17.5–44.8 h). There were 309 clinically suspected seizure episodes, of which 181 (58.6%) were the motor type and 128 (41.4%) were the non-motor type. Only 63 (20.4%) of these events were confirmed to be clinical seizures on a simultaneous video-EEG recording. In terms of the impact of continuous video-EEG monitoring on clinical management, the anti-epileptic drugs were changed in 42 (65.6%) of the 64 neonates. Conclusion: In the identification of neonatal seizures, a clinical diagnosis by direct observation alone is not enough. The use of continuous video-EEG monitoring plays an important role in the diagnosis of neonatal seizures and in guiding clinical management decisions.

## 1. Introduction

Neonatal seizures are a neonatal neurological emergency. In general clinical practice, neonatal seizures are identified by direct observation by staff. However, neonatal seizures are often subclinical, and abnormal clinical paroxysmal events may raise the suspicion of neonatal seizures [1]. Abnormal paroxysmal events include the motor type (abrupt, repetitive, or abnormal appearing movements) and non-motor type (atypical behavior or unprovoked episodes of autonomic dysfunction), and they may be the clinical manifestation of neonatal seizures [2,3]. Therefore, the identification of neonatal seizures is hampered by a variable clinical manifestation. Because it is difficult to decide clinically whether paroxysmal events are seizures, the incidence of neonatal seizures may be overestimated, resulting in unnecessary treatment [4].

Neonatal seizures may cause secondary brain injury and lead to long-term neurodevelopmental delay, and therefore prompt therapeutic interventions may be important [5,6]. The clinical observation of abnormal paroxysmal events without electroencephalography (EEG) confirmation has severe limitations, and therefore continuous video-EEG monitoring can provide an opportunity to differentiate epileptic seizures from non-epileptic events [7]. This study aimed to examine the frequency of clinical seizures in paroxysmal events in a neonatal intensive care unit (NICU). 

## 2. Materials and Methods

### 2.1. Study Population and Study Design 

We conducted a prospective study of continuous video-EEG monitoring in an NICU from January 2017 to December 2020. All neonates with clinical abnormal paroxysmal events and who underwent continuous video-EEG monitoring were enrolled (Figure 1). The clinical abnormal paroxysmal events were divided into two subgroups: “motor type” (clonic, subtle, tonic and myoclonic) and “non-motor type” (vital sign fluctuations) [2]. We excluded patients who: (1) were aged ≥48 weeks postmenstrual age (PMA); (2) received sedation and/or neuromuscular blockade during the recording; (3) did not have clinical abnormal paroxysmal events; and 4) had poor quality video-EEG recordings. This study was approved by the Ethical Committee of the Chang Gung Memorial Hospital Institutional Review Board (IRB numbers: 201600360B0 and 201901000B0). Written informed parental consent was obtained for each participant.

During the 4-year study period, 64 neonates who underwent continuous video-EEG monitoring and had clinical abnormal paroxysmal events were enrolled. We excluded 86 children, including 62 who did not have clinical abnormal paroxysmal events, 8 who received sedation and/or neuromuscular blockade during the recordings, 3 who were aged ≥48 weeks postmenstrual age (PMA), and 3 who had poor-quality video-EEGs. There were 309 suspected clinical seizure episodes, of which 181 (58.6%) were the motor type and 128 (41.4%) were the non-motor type. Only 63 (20.4%) of these events were confirmed to be electroclinical seizures in simultaneous video-EEG recordings (EEG: electroencephalography).

### 2.2. Continuous EEG Monitoring Protocol

A standardized continuous EEG monitoring protocol for neonates was initiated at Chang Gung Children’s Hospital in October 2016, according to consensus-based guidelines proposed in 2011 [8]. Bedside 16-channel continuous EEG monitoring was performed using a Nicolet Monitor (Natus Neuro, Middleton, WI, USA) video-EEG system, and neonatal EEG electrodes were placed (F4, C4, T4, O2, Fz, Cz, Pz, F3, C3, T3, O1). Electrocardiography and oxygen saturation were also measured simultaneously. The video-EEG recording began as soon as possible after recruitment. 

### 2.3. EEG Data Interpretation

EEG tracings were interpreted using standardized terminology of the American Clinical Neurophysiology Society (ACNS) for neonates [9]. Electrographic seizures were defined as EEG pattern that was evolving in frequency, distribution, or morphology, lasting > 10 s (or shorter if associated with a clinical change) [10,11]. EEG epochs without good video recordings for review were excluded from the analysis. 

### 2.4. Clinical Seizures Directly Observed by a Member of Staff

In our NICU setting, a bedside observer, including experienced nursing and medical staff, generally identifies clinical abnormal paroxysmal events, which they believe to be a clinical seizure. Nurses record a description of the event and the date and time, number and duration of each event electronically and/or on a bedside nursing sheet. In addition, they also push an EEG event button on the EEG monitoring system. Although the clinical staff are aware that the continuous video EEG monitoring studies are in progress, they are not trained in EEG reading.

The clinical abnormal paroxysmal events recorded after pushing the button on the continuous video-EEG recording system were later reviewed. A clinical seizure was defined as there being simultaneous electrographic seizures within 3 min before the recorded time of the clinically suspected seizure. The actual number of electroclinical seizures in the recordings made after pushing the button was calculated for each infant [4].

### 2.5. Data Collection 

We collected the following information from all patients: (1) demographics (sex, and gestational age at the start of recording); (2) monitoring indication, pre-existing diseases, and anti-epileptic drug administration during continuous video-EEG monitoring; (3) abnormal paroxysmal events, including abnormal movements and vital sign fluctuations; (4) continuous video-EEG findings; and (5) impact on clinical management. Preterm was defined as <37 weeks PMA and term as ≥37 weeks PMA in accordance with current neonatal EEG reporting standards [9]. Indications for monitoring were grouped based on established neonatal monitoring guidelines as (1) high-risk conditions and (2) suspected clinical seizures [8,9]. Pre-existing diseases included perinatal asphyxia, perinatal stroke and intraventricular hemorrhage, central nervous system infection, genetic disease/inborn error of metabolism (IEM)/hypocalcemia, prematurity, and others (periventricular leukomalacia and history of tetralogy of Fallot). 

### 2.6. Statistical Analysis

The patients were divided into the high-risk condition and suspected clinical seizure groups. The patients’ descriptive data are presented as mean ± standard deviation or median (interquartile range, IQR). Differences between groups were analyzed using the chi-square test or Fisher’s exact test for categorical variables, and the non-parametric Mann–Whitney test or Student’s parametric *t*-test for continuous variables. Statistical analysis was performed using SPSS software, version 23.0 (IBM, Inc., Chicago, IL, USA) with a significant value of *p* set at <0.05. 

## 3. Results

### 3.1. Patient Profile

During the 4-year study period, 150 neonates undergoing continuous video-EEG monitoring were identified, of whom 64 (42.6%) met the study entry criteria. Of the 86 children who were excluded, 62 did not have clinical abnormal paroxysmal events during the recording, 18 received sedation and/or neuromuscular blockade, 3 were aged ≥48 weeks PMA, and 3 had poor quality video-EEG recordings. At the time of starting continuous video-EEG monitoring, 45 (70.3%) of the patients were term infants. The gestational age at monitoring ranged from 27 6/7 to 45 3/7 weeks (median 38.0 weeks, IQR 35.3–40.0 weeks). In terms of the indication for continuous EEG monitoring, 14 (21.9%) patients were monitored for high-risk conditions and 50 (78.1%) were monitored for suspected clinical seizures. The patients’ characteristics are summarized in Table 1.

### 3.2. EEG Recording and Findings

The median total EEG monitoring duration was 24.1 h (IQR 17.5–44.8 h), including 51.9 h (IQR 31.4–76.7 h) in the high-risk conditions group and 21.9 h (IQR 17.3–42.2 h) in the suspected clinical seizures group. Clinical seizures occurred in 22 (34.4%) of the 64 neonates. Clinically suspected seizures without electrographic evidence occurred in 42 (65.6%) neonates. 

### 3.3. Continuous Video-EEG Monitoring of Clinical Abnormal Paroxysmal Events

In the 64 enrolled neonates, there were 309 clinically suspected seizure episodes. Of these episodes, 181 (58.6%) were the motor type and 128 (41.4%) were the non-motor type. Only 63 (20.4%) of these events documented in medical/nursing notes were confirmed to be electrographic seizures on simultaneous video-EEG recordings. The most common motor type was muscle clonus/jitteriness (77 of 181, 42.5%), followed by subtle movements such as eye blinking, mouthing, and fisting (64 of 181, 35.4%). However, the diagnostic accuracies of these two types for clinical seizures were only 31.2% and 17.2%, respectively. The diagnostic accuracies of muscle clonus/jitteriness, tonic, and myoclonic movement for clinical seizures were more common in term, and the diagnostic accuracies of subtle movements were more common in preterm. In addition, the diagnostic accuracy of the non-motor type for clinical seizures was lower than that for the motor type (0.8%, 1 of 128 episodes). The types and diagnostic accuracy of the clinical paroxysmal events and clinical seizures are summarized in Table 2. 

### 3.4. Impact of Continuous Video-EEG on Clinical Management 

In this study, we enrolled all neonates with clinical abnormal paroxysmal events, which staff believed to be a clinical seizure. Fifty-two (81.2%) of the neonates had received an anti-epileptic drug therapy by clinical observation during the continuous video-EEG monitoring. After review, continuous video-EEG monitoring led to a change in clinical management in 65.6% of the neonates (42 of 64), including initiating anti-epileptic drugs (1, 1.6%), anti-epileptic drug escalation (16, 25%), and anti-epileptic drug discontinuation (25, 39.1%) (Figure 2). At discharge, in terms of 42 infants who had anti-epileptic drugs changed because of continuous EEG recordings, 5 children died, 17 children received 1 anti-epileptic drug, 6 received more than 2 anti-epileptic drugs, and 14 children did not receive any anti-epileptic drug. In terms of the 22 infants who did not have their anti-epileptic drugs changed, 11 children received one anti-epileptic drug and 11 children do not receive any anti-epileptic drug. 

## 4. Discussion

The clinical diagnosis of neonatal seizures by direct observation alone without EEG monitoring is a challenge. In this study, we determined the frequency of clinical seizures in paroxysmal events in an NICU. We found that most of the clinically paroxysmal events were misinterpreted as neonatal seizures by experienced neonatal staff. Moreover, continuous video-EEG monitoring led to a change in clinical management in parts of the neonates. Therefore, diagnosing neonatal seizures by direct observation alone is not enough.

The most common abnormal movements misinterpreted as clinical seizures were generalized muscle clonus/jitteriness and subtle movements such as mouthing and fisting. In our study, the diagnostic accuracy of these two types of abnormal movements for clinical seizure was low at 31.2% and 17.2%, respectively. Thus, neonatal seizures are difficult or impossible to accurately identify by direct observation alone.

In clinical practice, due to developing neurophysiology and immature movements, neonates often present with paroxysmal movements and vital sign fluctuations. However, these clinical signs may be also the clinical manifestation of neonatal seizures. Because it is difficult to identify neonatal seizures, the incidence of clinical seizures may be overestimated [4]. Murray et al. conducted a study to examine the frequency of neonatal seizures identified by direct observation and concluded that only 48 (27%) of the 177 clinically suspected seizure episodes were confirmed to be electrographic seizures [4]. Malone et al. conducted a study to determine the interobserver agreement in neonatal seizure identification when presented with clinical histories and digital video recordings only. The average number of correctly identified events was 50%. Clonic seizures were correctly identified most frequently, and subtle seizures were poorly identified. Moreover, the interobserver agreement (Kappa) for doctors and other health care professionals was also poor at 0.21 and 0.29, respectively [12]. In our study, of 309 episodes, 181 (58.6%) were the motor type and 128 (41.4%) were the non-motor type. Only 63 (20.4%) of these events documented in medical/nursing notes were confirmed to be electrographic seizures on simultaneous video-EEG recording. Therefore, continuous video-EEG monitoring plays an important role in differentiating paroxysmal non-epileptic events from clinical seizures in neonates [7].

The over-diagnosis of neonatal seizures by experienced neonatal staff is also common, which can result in unnecessary and potentially harmful treatment. Continuous video-EEG monitoring allows for the prompt treatment of electrographic seizures [13,14]. In Murray et al.’s study, 9 of the 51 neonates with suspected clinical seizures were found to have electrographic seizures [4]. In the present study, we found that continuous video-EEG monitoring led to a change in clinical management in parts of the neonates, as a result of demonstrating that events were not seizures as well as seizure detection and management. Therefore, anti-epileptic drugs and their associated potential adverse effects can be avoided or decreased. Our results emphasize that the use of continuous video-EEG monitoring plays an important role in helping to limit misdiagnosis and overtreatment in an NICU.

## 5. Limitations

This study has several limitations. First, routine clinical practice at our institution is to initiate or adjust anticonvulsants in neonates when clinical seizures are detected. In this study, we enrolled all neonates with clinical abnormal paroxysmal events that members of staff believed to be a clinical seizure. Although most (81.2%) neonates had received an anti-epileptic drug during the continuous video-EEG monitoring, variations in clinical practice with regard to initiating or adjusting anti-epileptic drug treatment were observed in this study. In addition, anti-epileptic drug therapy often leads to electroclinical dissociation, meaning that the number of clinical seizures decreases while the number of electrographic seizures continues unabated or increases [15,16]. Thus, neonatal seizures are difficult or impossible to accurately identify by direct observation alone. Second, the time of EEG interpretation was variable. In addition, neonatologists were not involved in EEG interpretation in this study, and EEG interpretation was performed by three electroencephalographers (Lin, J.J; Wang, Y.S.; and Kuo, C.Y.). We recognize that a wide range of clinical circumstances influence EEG review practices and treatment strategies as dictated by institutional resources. Third, since the impact of the continuous video EEG-related management changes on the outcome is unknown, further prospective studies are needed to determine whether detecting and treating electrographic seizures prompted by continuous EEG can improves outcome.

## 6. Conclusions

In conclusion, this study demonstrated that most clinical paroxysmal events were misinterpreted as neonatal seizures by experienced neonatal staff. Additionally, continuous video-EEG monitoring led to changes in clinical management in parts of the neonates who underwent continuous video-EEG. These findings suggest that in the identification and management of neonatal seizures, making a clinical diagnosis by direct observation alone is not enough. The use of continuous video-EEG monitoring provides an opportunity to detect whether a paroxysmal event is a clinical seizure, and continuous video-EEG results can also assist in making important treatment decisions.

## Figures and Tables

**Figure 1 children-09-00238-f001:**
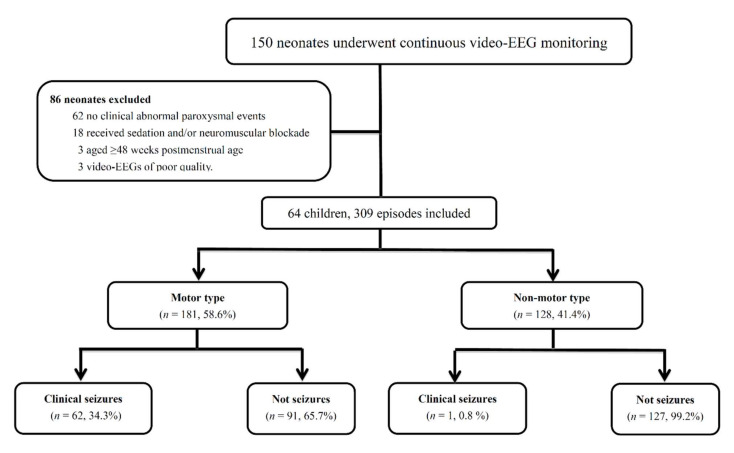
Flowchart of the study.

**Figure 2 children-09-00238-f002:**
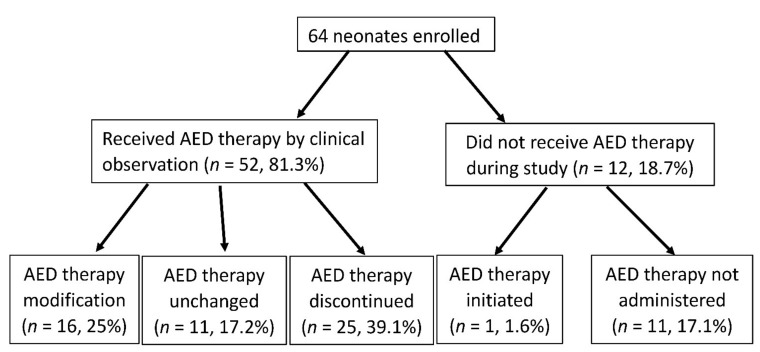
Impact of continuous video-EEG monitoring on clinical management. Fifty-two (81.2%) neonates had received an anti-epileptic drug therapy by clinical observation during continuous video-EEG monitoring. Finally, continuous video-EEG monitoring led to a change in anti-epileptic drug treatment in 65.6% of neonates (42 of 64). (AED: anti-epileptic drug).

**Table 1 children-09-00238-t001:** Patient demographics and baseline characteristics by indication.

	All (*n* = 64)	High-Risk Conditions (*n* = 14)	Suspected Clinical Seizures (*n* = 50)	*p* Value
Sex				1.000
Male	50 (78.1%)	11 (78.6%)	39 (78%)	
Female	14 (21.9%)	3 (21.4%)	11 (22%)	
Age				
Gestational age at start of recording (weeks)	37.40 + 4.26	37.78 ± 2.54	37.3 ± 4.64	0.611
Preterm ^a^	19 (29.7%)	4 (28.6%)	15 (30%)	1.000
Term ^b^	45 (70.3%)	10 (71.4%)	35 (70%)	
Pre-existing disease				
Previously healthy	17 (26.6%)	0 (0%)	17 (34%)	
Perinatal asphyxia	23 (35.9%)	13 (92.9%)	10 (20%)	
Perinatal stroke	5 (7.8%)	0 (0%)	5 (10%)	
CNS infection	5 (7.8%)	1 (7.1%)	4 (8%)	
Genetic disease/IEM/hypocalcemia	6 (9.4%)	0 (0%)	6 (12%)	
Premature infant	6 (9.4%)	0 (0%)	6 (12%)	
Other	2 (3.1%)	0 (0%)	2 (4%)	
cvEEG recording and finding				
Duration of cvEEG recording (hours)	24.1 (IQR: 17.5–44.8)	51.9 (IQR: 31.4–76.7)	21.9 (IQR: 17.3–42.2)	0.004 *
Clinical seizures (frequency)	2.14 + 4.91	4.71 + 7.59	1.42 + 3.64	0.046 *
Clinical management				
Treatment with AED during cvEEG	52 (81.3%)	14 (100%)	38 (76%)	0.054

Abbreviations: CNS: central nervous system; cvEEG: continuous video-EEG monitoring; AED: anti-epileptic drug; IEM: inborn errors of metabolism; * *p* < 0.05: statistically significant; ^a^ Preterm: <37 weeks; ^b^ Term: ≥37weeks.

**Table 2 children-09-00238-t002:** The type and diagnostic accuracy of clinical paroxysmal events for clinical seizures.

Clinical Events	Number	Clinical Seizures on cvEEG Monitoring (%)	Clinical Seizures on Term Baby (%)	Clinical Seizures on Preterm Baby (%)
Motor type	181	62/181 (34.3%)	52/125 (41.6%)	10/56 (17.8%)
Muscle clonus/jitteriness	77	24/77 (31.2%)	24/49 (48.8%)	0/28 (0%)
Subtle movements	64	11/64 (17.2%)	5/48 (10.4%)	6/16 (37.5%)
Tonic	30	18/30 (60%)	14/18 (77.8%)	4/12 (33.3%)
Myoclonic	10	9/10 (90%)	9/10 (90%)	0/0 (-)
Non-motor type				
Vital sign fluctuations	128	1/128 (0.8%)	1/114 (0.8%)	0/14 (0%)

cvEEG: continuous video-EEG monitoring.

## Data Availability

The data presented in this study are available on request from the corresponding author. The data are not publicly available due to privacy and ethical issue.

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
