# Peer review of "The Frequency of Clinical Seizures in Paroxysmal Events in a Neonatal Intensive Care Unit"

_children, 2022, doi:10.3390/children9020238_

Round 1

Reviewer 1 Report

Significant study limitations which can affect the generalizability of the study. Majority of enrolled patients already on AED rx – this could adversely affect true correlation of movements with EEG. In addition, variability in time to read EEG concerning. A statement about these and adjusting the conclusion to reflect that one cannot clearly draw the conclusion reached  is very important.

Author Response

Reviewer 1 comment:

Significant study limitations which can affect the generalizability of the study. Majority of enrolled patients already on AED rx – this could adversely affect true correlation of movements with EEG.

Ans: Thanks for your comment. We total agreed your comment that anti-epileptic drug therapy often leads to electroclinical dissociation, meaning that the number of clinical seizures decreases while the number of electrographic seizures continues unabated or increases [Boylan GB, Rennie JM, Pressler RM, Wilson G, Morton M, Binnie CD. Phenobarbitone, neonatal seizures, and video-EEG. Arch Dis Child Fetal Neonatal Ed 2002;86:F165-170. Scher MS, Alvin J, Gaus L, Minnigh B, Painter MJ. Uncoupling of EEG-clinical neonatal seizures after antiepileptic drug use. Pediatr Neurol. 2003;28:277-280.] Because in general clinical practice, neonatal seizures are identified by direct observation by a staff, this study aimed only to examine the frequency of clinical seizures in paroxysmal events in a neonatal intensive care unit in the real world. We don’t focus electrographic seizures in these patients. Therefore, we added the above description in the limitation in the page 7 line 226 to line 228.

In addition, variability in time to read EEG concerning.

Ans: Thanks for your important comments. In the real world, the time of EEG interpretation was variable. We recognize that a wide range of clinical circumstances influence EEG review practices and treatment strategies as dictated by institutional resources. And we have mentioned this limitation in the limitation section in the page 7 line 229 to line 233.

A statement about these and adjusting the conclusion to reflect that one cannot clearly draw the conclusion reached is very important.

Ans: Thanks for your comments. And we revised the conclusion in the page 7 line 239 to line 242 as following:

“In conclusion, this study demonstrated that most of clinical paroxysmal events were misinterpreted as neonatal seizures by experienced neonatal staff. And continuous video-EEG monitoring led to changes in clinical management in parts of the neonates who underwent continuous video-EEG.”

Reviewer 2 Report

This report is very important from the perspective of preventing inappropriate administration of anticonvulsants for neonates. In clinical practice, it is difficult to apply all neonates continuous video-EEG monitoring. Were there any observational differences between seizures confirmed by EEG and those not? Feedback of EEG findings may improve observation skills of neonatal episodes.  

Author Response

Reviewer 2

This report is very important from the perspective of preventing inappropriate administration of anticonvulsants for neonates. In clinical practice, it is difficult to apply all neonates continuous video-EEG monitoring. Were there any observational differences between seizures confirmed by EEG and those not? Feedback of EEG findings may improve observation skills of neonatal episodes.  

Ans: Thanks for your important comment. As your comment, it is difficult to apply all neonates continuous video-EEG monitoring in real world. In our study, we try to analyze the observational differences between seizures confirmed by EEG and those not. And we have described the results in the result section in the page 5 line 148 to line 161and table 2 as following:

3.3. Continuous video-EEG monitoring of clinical abnormal paroxysmal events

In the 64 enrolled neonates, there were 309 clinically suspected seizure episodes. Of these episodes, 181 (58.6%) were the motor type and 128 (41.4%) were the non-motor type. Only 63 (20.4%) of these events documented in medical/nursing notes were confirmed to be electrographic seizures on simultaneous video-EEG recordings. The most common motor type was muscle clonus/jitteriness (77 of 181, 42.5%), followed by subtle movements such as eye blinking, mouthing, and fisting (64 of 181, 35.4%). However, the diagnostic accuracies of these two types for clinical seizures were only 31.2% and 17.2%, respectively. The diagnostic accuracies of muscle clonus/jitteriness, tonic and myoclonic movement for clinical seizures were more common in term and the diagnostic accuracies of subtle movements was more common in preterm. In addition, the diagnostic accuracy of the non-motor type for clinical seizures was lower than that for the motor type (0.8%, 1 of 128 episodes). The types and diagnostic accuracy of the clinical paroxysmal events and clinical seizures are summarized in Table 2.

Table 2. The type and diagnostic accuracy of clinical paroxysmal events for clinical seizures.

Clinical events

Number

Clinical seizures on cvEEG monitoring (%)

Clinical seizures on Term baby (%)

Clinical seizures on Preterm baby (%)

Motor type

181

62/181 (34.3%)

52/125 (41.6%)

10/56 (17.8%)

 Muscle clonus/jitteriness

77

24/77 (31.2%)

24/49 (48.8%)

0/28 (0%)

 Subtle movements

64

11/64 (17.2%)

5/48 (10.4%)

6/16 (37.5%)

 Tonic

30

18/30 (60%)

14/18 (77.8%)

4/12 (33.3%)

 Myoclonic

10

9/10 (90%)

9/10 (90%)

0/0 (-)

Non-motor type

Vital sign fluctuations

128

1/128 (0.8%)

1/114 (0.8%)

0/14 (0%)

cvEEG: continuous video-EEG monitoring.
